# Rethinking Distance Metric Generalization in Neural Combinatorial Optimization for Vehicle Routing Problems

## Abstract

Neural combinatorial optimization (NCO) has emerged as a promising approach for solving the vehicle routing problem (VRP). However, its ability to generalize across diverse instances is a key challenge for practical applications. Current research on generalization primarily focuses on problem scale and node distribution. The distance metric between nodes, such as 2D Euclidean distance or geographical distance, is also an important characteristic of VRP instances. Unfortunately, existing NCO methods typically use a single distance metric for both training and testing, neglecting the diversity of distance metrics. To fill this gap, this paper systematically investigates the impact of distance metrics. First, we introduce a benchmarking framework that supports multiple distance metrics and evaluates model generalization across them. Experimental results reveal that models trained on instances with a single distance metric perform poorly on instances with different metrics. This suggests that variations in distance metrics pose a significant challenge to model generalization. Second, we examine several training data configurations and find that jointly training on data with diverse distance metrics significantly improves model generalization across different metrics. Moreover, by integrating our proposed method for distance metric generalization with prior advances for problem scale and node distribution generalization, the performance of NCO models on various real-world VRP instances is substantially improved.

## 1 Introduction

The vehicle routing problem (VRP) (Toth & Vigo, 2002; 2014) aims to assign routes to vehicles so that all nodes are visited and the total length of the route is minimized. The VRP has a wide range of practical applications, including logistics (Veres & Moussa, 2019), transportation (Garaix et al., 2010), and ride-hailing services (Laporte, 2009). However, obtaining optimal VRP solutions is extremely difficult due to the NP-hardness of the problem (Ausiello et al., 2012; Papadimitriou & Steiglitz, 1998). Traditional approximate algorithms (e.g., LKH3 (Helsgaun, 2017) and HGS (Vidal, 2022)) solve the VRP using carefully crafted heuristics, and developing them requires substantial domain expertise. In recent years, neural combinational optimization (NCO) methods have emerged as a promising alternative paradigm for solving the VRP (Bengio et al., 2021). By leveraging neural networks, NCO methods automatically learn effective solving policies from data and produce high-quality solutions without relying on domain expertise (Kwon et al., 2020; Ma et al., 2021; Sun et al., 2024; Ye et al., 2024; Zheng et al., 2024; Luo et al., 2025; Zhou et al., 2025).

Generalization is a key challenge to the practical deployment of NCO methods. This is because real-world VRPs encompass diverse scenarios, each characterized by distinct data characteristics. Current research on generalization primarily focuses on two aspects: problem scale (i.e., number of nodes) (Luo et al., 2023; Drakulic et al., 2023; Ye et al., 2024; Fang et al., 2024; Zheng et al., 2024; Zhou et al., 2025) and node distribution (e.g., uniform distribution, clustered distribution) (Jiang et al., 2022; Zhou et al., 2023; Bi et al., 2022). However, we notice that the distance metric (e.g., 2D Euclidean distance, 3D Euclidean distance, geographical distance,and real-world road network distance) is also a diverse aspect in real-world VRPs. For instance, for the application of VRP [1],

---

[1] https://www.math.uwaterloo.ca/tsp/world/

visiting all locations registered as populated places around the world, distance computations between nodes are based on geographical distance. While for other applications [2], such as those involving star-to-star travel, the appropriate metric is the 3D Euclidean distance.

Unfortunately, research on the generalization across distance metrics (i.e., distance metric generalization) remains limited. Most existing methods (Kwon et al., 2020; Drakulic et al., 2023; Luo et al., 2023) use 2D Cartesian coordinates as model inputs and usually adopt the 2D Euclidean distance metric for both training and testing, neglecting the diversity of distance metrics. Besides, this coordinate-based input format inherently couples the model with some specific distance metrics, limiting its applicability to other distance metrics. For example, models that take 2D Cartesian coordinates as input cannot be directly applied to real-world logistics problems that require the geographical distance metric, which relies on geographical coordinates like latitude and longitude. These models, adopting the coordinate-based input format, are fundamentally limited in supporting various distance metrics, let alone generalizing across them. This oversight hinders the widespread application of NCO methods in real-world VRPs.

This paper presents the first systematic empirical study on distance metric generalization. A benchmarking framework is proposed, and three experimental studies are conducted. In the experiments, the distance matrix, a square matrix where each element represents the distance between two nodes, serves as the model input (Kwon et al., 2021; Drakulic et al., 2024; Pan et al., 2025), since this input format decouples the model from specific coordinate systems and allows it to naturally support any distance metric. Our experimental findings underscore the importance of distance metric generalization and provide guidance for developing universal neural solvers. Our contributions are summarized as follows:

- We propose a benchmarking framework to evaluate model generalization across distance metrics. This framework supports eight distance metrics and can generate instances according to different metrics for standardized evaluation.

- We reveal that changes in distance metrics pose significant challenges to model generalization. In our first experiment, the models are trained on instances with a single distance metric, which is a common practice in NCO methods. These models are prone to overfitting to the single metric and have inferior performance on instances with other metrics.

- We propose the Multi-Metric training, a simple yet efficient method. The second experiment investigates various training configurations with respect to the distance metric. The Multi-Metric training refers to jointly training on instances that involve multiple distance metrics. Our findings indicate that the Multi-Metric training effectively enhances model generalization across distance metrics.

- Using the Multi-Metric training can achieve substantial performance gains on real-world VRPs. The final experiment integrates the Multi-Metric training with recent advances in generalization across problem scales and node distributions. The Multi-Metric training further improves the performance on real-world datasets.

## 2 RELATED WORK

### 2.1 NCO MODEL INPUT FORMATS

**Coordinate-Based** The coordinate-based input format is the predominant input format in NCO for solving the VRP. In this format, the input to the model is typically the coordinates of the nodes in the problem instance. The model then learns to optimize the VRP based on these spatial coordinates. Vinyals et al. (2015), Bello et al. (2016), and Nazari et al. (2018) leverage recurrent neural networks (RNNs) to solve VRPs. Kool et al. (2018) and Deudon et al. (2018) boost the performance by introducing the Transformer architecture (Vaswani et al., 2017). Subsequently, numerous improved versions of the Transformer architecture have been proposed (Nazari et al., 2018; Kool et al., 2018; Deudon et al., 2018; Xin et al., 2021; 2020; Kwon et al., 2020; Hottung et al., 2021; Kim et al., 2021; 2022; Jiang et al., 2023; Sun et al., 2024; Zhou et al., 2024; Luo et al., 2025). Furthermore, some models (Qiu et al., 2022; Sun & Yang, 2023) take both node coordinates and distance between nodes as inputs.

---

[2] https://www.math.uwaterloo.ca/tsp/star/about.html

Although the methods based on the coordinate-based input format have shown promising results, they are inherently limited by their inability to support arbitrary distance metrics. This restriction reduces their applicability in real-world VRPs that involve diverse distance metrics. The limitation arises from the reliance on a specific coordinate system, which confines the method to distance metrics compatible with that particular coordinate system. For instance, models based on 2D Cartesian coordinates cannot be directly applied to scenarios that require calculating geographical distances, which relies on geographical coordinates like latitude and longitude.

**Distance-Matrix-Based**    In the distance-matrix-based input format, the input to the model is a distance matrix that contains the pairwise distances between nodes. Several notable works have explored NCO models utilizing this format, including Kwon et al. (2021), Drakulic et al. (2024), Pan et al. (2025), Lischka et al. (2024), and Meng et al. (2025).

Each element in the distance matrix encodes the distance between two nodes. These distances can be computed through any distance metric. Therefore, the distance-matrix-based input format supports arbitrary distance metrics and thus is a more general format for NCO methods.

## 2.2    Generalization Challenges of NCO Methods in VRPs

**Problem Scales**    Generalization across problem scales refers to the model's ability to perform well on instances of varying numbers of nodes. Some methods (Luo et al., 2023; Drakulic et al., 2023) modify the model architecture to enhance generalization. Others (Li et al., 2021; Manchanda et al., 2022; Gao et al., 2023; Zhou et al., 2024) employ training data of varying scales to improve generalization performance. In addition, several studies (Luo et al., 2023; Ye et al., 2024; Zheng et al., 2024) decompose problems into sub-problems, and then solve these sub-problems individually.

**Node Distributions**    In many practical scenarios, node distributions in VRP instances can differ substantially, which may affect the performance of the model. Generalization across node distributions refers to the model's ability to perform well regardless of how the nodes are distributed. In general, recent research on generalization across node distributions primarily trains models from data across diverse distributions to learn a universal solving policy. Representative works include Jiang et al. (2022), Bi et al. (2022), and Zhou et al. (2023).

**Distance Metrics**    In many real-world VRP instances, distance metrics between nodes are different. To ensure broad applicability across diverse real-world scenarios, it is required that NCO models can handle instances with different distance metrics. Recent efforts have attempted to adapt models to instances with different distance metrics. Among them, Pan et al. (2025) applies the model to TSPLIB (Reinelt, 1991) instances with four distinct distance metrics. Lischka et al. (2024) consider instances with symmetric Euclidean distances, asymmetric distances that are randomly sampled and adhere to the triangle inequality, and asymmetric distances that are randomly sampled and violate the triangle inequality. However, these studies lack systematic testing and comprehensive analysis of model generalization across distance metrics. Our work revisits the challenge of the distance metric generalization and proposes a benchmarking framework to systematically analyze and evaluate the generalization ability of NCO methods under varying distance metrics.

## 3    Testing Distance Metric Generalization of NCO Methods

Existing NCO methods for the VRP typically train a model on data with the 2D Euclidean distance metric. This section uses the traveling salesman problem (TSP) as a case study to evaluate the generalization performance of models trained under this configuration on instances with different distance metrics.

## 3.1 Distance Metrics for Benchmarking

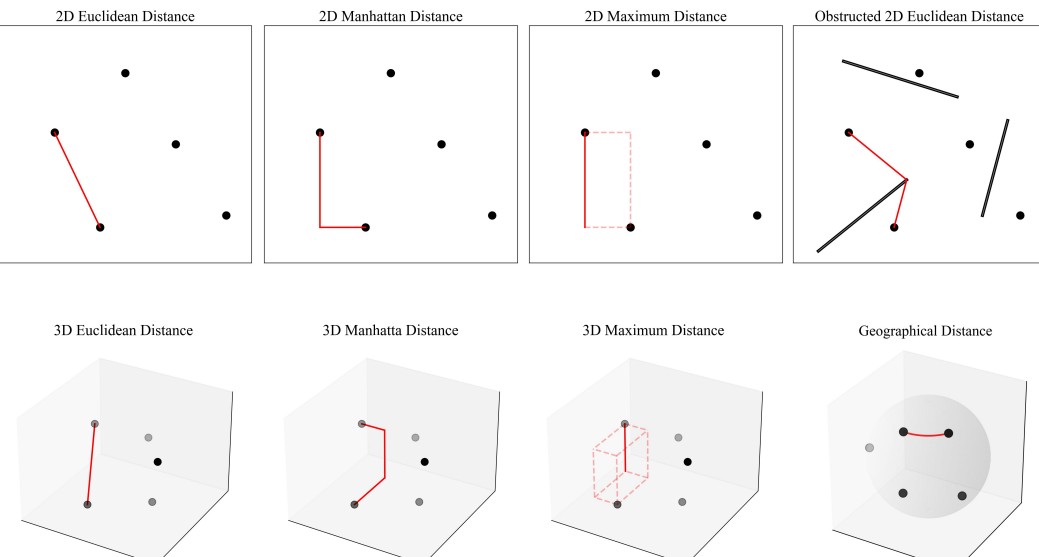

Figure 1: Visualization of the eight distance metrics implemented in the benchmarking framework. The red line represents the shortest path between nodes under different distance metrics.

To evaluate the model's generalization to instances of different distance metrics, we construct a benchmarking framework. This benchmarking framework covers multiple distance metrics, including 2D Euclidean distance, 3D Euclidean distance, 2D Manhattan distance, 3D Manhattan distance, 2D Maximum distance, 3D Maximum distance, and geographical distances. These distance metrics are illustrated in Figure 1.

**2D Euclidean Distance**  The 2D Euclidean distance measures the straight-line distance between two nodes in a plane. Each node is defined by 2D Cartesian coordinates $(x, y)$. The distance between two nodes $p = (x_1, y_1)$ and $q = (x_2, y_2)$ is computed as:

$$d(p, q) = \sqrt{(x_2 - x_1)^2 + (y_2 - y_1)^2}. \tag{1}$$

**3D Euclidean Distance**  This metric extends the 2D Euclidean distance to three-dimensional space. Nodes are represented by 3D Cartesian coordinates $(x, y, z)$. The distance between two nodes $p = (x_1, y_1, z_1)$ and $q = (x_2, y_2, z_2)$ is given by:

$$d(p, q) = \sqrt{(x_2 - x_1)^2 + (y_2 - y_1)^2 + (z_2 - z_1)^2}. \tag{2}$$

**2D Manhattan Distance**  The 2D Manhattan distance is the sum of the absolute differences of the 2D Cartesian coordinates of two nodes. Each node is represented as $(x, y)$. The distance between nodes $p = (x_1, y_1)$ and $q = (x_2, y_2)$ is calculated using the following formula:

$$d(p, q) = |x_2 - x_1| + |y_2 - y_1|. \tag{3}$$

**3D Manhattan Distance**  The 3D Manhattan distance is an extension of the 2D case, applied to three-dimensional space. Nodes are defined by 3D Cartesian coordinates $(x, y, z)$. The distance between two nodes $p = (x_1, y_1, z_1)$ and $q = (x_2, y_2, z_2)$ is:

$$d(p, q) = |x_2 - x_1| + |y_2 - y_1| + |z_2 - z_1|. \tag{4}$$

**2D Maximum Distance**   The 2D Maximum distance is defined as the maximum absolute difference between two nodes in a plane. Nodes are represented as 2D Cartesian coordinates $(x, y)$. The distance between node $p = (x_1, y_1)$ and node $q = (x_2, y_2)$ is:

$$d(p, q) = \max\left(|x_2 - x_1|, |y_2 - y_1|\right). \tag{5}$$

**3D Maximum Distance**   This extends the 2D Maximum distance to three-dimensional space. Nodes are defined by 3D Cartesian coordinates $(x, y, z)$. The distance between two nodes $p = (x_1, y_1, z_1)$ and $q = (x_2, y_2, z_2)$ is computed as:

$$d(p, q) = \max\left(|x_2 - x_1|, |y_2 - y_1|, |z_2 - z_1|\right). \tag{6}$$

**Geographical Distance**   The geographical distance measures the distance between two nodes on the Earth's surface, where the Earth is treated as an idealized sphere with a radius of 6378.388 kilometers. Nodes are specified by their latitude and longitude coordinates $(\phi, \lambda)$. The distance between two nodes $p = (\phi_1, \lambda_1)$ and $q = (\phi_2, \lambda_2)$ is given by:

$$d(p, q) = R \cdot \arccos\left(\sin\phi_1 \sin\phi_2 + \cos\phi_1 \cos\phi_2 \cos(\lambda_1 - \lambda_2)\right) \tag{7}$$

where $R$ denotes the Earth's radius.

**Obstructed 2D Euclidean Distance**   The obstructed 2D Euclidean distance extends the standard 2D Euclidean distance by accounting for obstacles that block straight-line paths between nodes. Each node is defined by 2D Cartesian coordinates. When an obstacle intersects the straight-line path two nodes, the path between these two nodes must navigate around the obstacle. For two nodes $p$ and $q$, the distance $d(p, q)$ is defined as the length of the shortest path between them that avoids all obstacles. Although there is no explicit mathematical formula for this distance metric, the distance between nodes can be computed using algorithms that identify the shortest path while avoiding all obstacles. Specifically, distance matrices of TSP instances with obstacles are computed by applying Dijkstra's algorithm (DIJKSTRA, 1959) to the visibility graph. The visibility graph is constructed by including all nodes and obstacle endpoints as points, with edges connecting any two points that are mutually visible (i.e., the straight-line path between them does not intersect any obstacle). Then, Dijkstra's algorithm is used to compute the shortest path between each pair of nodes in this graph, and lengths of these path populate the entries of the distance matrix.

### 3.2   Experimental Setups

**Test Dataset**   The test dataset comprises instances with the eight distance metrics described in Section 3.1, with each distance metric containing 10,000 instances of TSP100 (i.e., TSP instances with 100 nodes). For distance metrics based on 2D Cartesian coordinates, coordinates are uniformly sampled within a unit square. For distance metrics based on 3D Cartesian coordinates, coordinates are uniformly sampled within a unit cube. For metrics based on latitudelongitude coordinates, coordinates are uniformly sampled within the feasible ranges (i.e., longitude: $[-180, 180]$, latitude: $[-90, 90]$). The setting of instances with obstructed 2D Euclidean distance is based on those described in VanDrunen et al. (2023). The obstacles are represented as line segments. The midpoints of these obstacles are uniformly sampled within the unit square, and their orientations are also uniformly sampled. Nodes cannot be fully enclosed by obstacles in such a way that makes them inaccessible. If a node is enclosed, the instance will be regenerated. Each TSP100 instance consists of 20 obstacles, each with a length of 0.25. Instances are represented as distance matrices, derived from the respective distance metrics. All distance matrices are normalized such that every value lies in the interval $[0, 1]$.

**Model & Training**   We conduct our experiments on the MatNet (Kwon et al., 2021), which adopts the distance-matrix-based input format. This model is trained via reinforcement learning and requires no additional labels. We train the model on TSP100 instances with the 2D Euclidean distance. The node coordinates are uniformly sampled within a unit square. Instances are represented as distance matrices, and all distance matrices are normalized. In addition, we train the model for

5,500 epochs, with 10,000 samples per epoch and a batch size of 100. The model hyperparameters, other training hyperparameters, and the inference strategy are identical to those in the original paper. All experiments are executed on an Intel(R) Xeon(R) Gold 6348 CPU or a Tesla V100-PCIE-32GB GPU.

## 3.3 RESULTS AND ANALYSIS

| Distance metric | Concorde (Applegate et al., 2006) | | MatNet (Kwon et al., 2021) | |
|---|---|---|---|---|
| | Obj. | Gap | Obj. | Gap |
| 2D Euclidean Distance | 6.0866 | 0.00% | 6.2100 | 2.03% |
| 3D Euclidean Distance | 11.6972 | 0.00% | 19.5522 | 67.15% |
| 2D Manhattan Distance | 5.3755 | 0.00% | 6.1679 | 14.74% |
| 3D Manhattan Distance | 9.8901 | 0.00% | 16.5675 | 67.52% |
| 2D Maximum Distance | 7.0107 | 0.00% | 8.5595 | 22.09% |
| 3D Maximum Distance | 13.9553 | 0.00% | 26.9987 | 93.47% |
| Geographical Distance | 7.7222 | 0.00% | 14.5277 | 88.13% |
| Obstructed 2D Euclidean Distance | 6.5590 | 0.00% | 7.3834 | 12.57% |

Table 1: Performance of the model on instances with multiple distance metrics, where the model is trained on instances with 2D Euclidean metrics. Average objective value (Obj.) indicates the average solution length. Gap measures the gap of objective value to Concorde (Applegate et al., 2006).

As shown in Table 1, the model achieves low gaps when tested on instances with the same distance metric as used in training (i.e., the Gap on the instances with 2D Euclidean distance is 2.03%). However, when the test instances with an unseen distance metric (i.e., a distance metric that differs from the one used during training), the model performance declines significantly. A minimum of a 6-fold improvement in the gap is observed across all instances with unseen distance metrics. In the most extreme case, the gap increases by 46-fold (i.e., the gap on instances with the 3D Maximum distance is 93.47%, which is 46 times larger than the gap on instances with the 2D Euclidean distance). These results demonstrate that the training configurations typically used in current NCO methods (i.e., training on instances with the 2D Euclidean distance) cause the model to overfit a specific distance metric, leading to poor performance when applied to other metrics.

## 4 IMPROVING DISTANCE METRIC GENERALIZATION FOR NCO

Training a model on data with only a single distance metric can induce overfitting to that specific metric, thereby leading to performance degradation when evaluated with other metrics. In this section, we examine the effects of different training configurations on the distance metric generalization of NCO models.

### 4.1 EXPERIMENTAL SETUPS

**Training Data Configurations**   We adopt three distinct training data configurations.

- Single-Metric Training (SMT): This configuration is identical to the training configuration in Section 3.2, where the model is trained solely on instances with the 2D Euclidean distance.
- Random: Inspired by the setup in MatNet (Kwon et al., 2021), the distances between nodes (i.e., the values in the distance matrix) are randomly sampled without adhering to a specific distance metric. Additionally, the diagonal elements of the distance matrix are set to zero, and the matrix is reconstructed to satisfy both symmetry and the triangle inequality.

- Multi-Metric Training (MMT): Under this configuration, training instances incorporate three different distance metrics: 2D Euclidean, 3D Euclidean, and Geographical distance. During training, instances with one distance metric are used per epoch. The three distance metrics rotate sequentially across epochs.

All training configurations are conducted on TSP100 instances with the total number of training instances kept constant. For the three distance metrics (2D Euclidean, 3D Euclidean, and geographical distance), the node distributions are consistent with the test datasets described in Section 3.2. All instances are represented as distance matrices and distance matrices are normalized. Except for the training data configurations described above, all other experiment settings (e.g., test datasets, training hyperparameters, model hyperparameters, and inference strategy) remain identical to those described in Section 3.2.

## 4.2 RESULTS AND ANALYSIS

| Distance metric | Concorde | | MatNet-SMT | | MatNet-Random | | MatNet-MMT | |
|---|---|---|---|---|---|---|---|---|
| | Obj. | Gap | Obj. | Gap | Obj. | Gap | Obj. | Gap |
| 2D Euclidean Distance | 6.0866 | 0.00% | 6.2100 | 2.03% | 13.6988 | 125.06% | 6.2105 | 2.03% |
| 3D Euclidean Distance | 11.6972 | 0.00% | 19.5522 | 67.15% | 13.9523 | 19.28% | 11.9722 | 2.35% |
| 2D Manhattan Distance | 5.3755 | 0.00% | 6.1679 | 14.74% | 13.4617 | 150.43% | 5.7398 | 6.78% |
| 3D Manhattan Distance | 9.8901 | 0.00% | 16.5675 | 67.52% | 13.5525 | 37.03% | 10.8772 | 9.98% |
| 2D Maximum Distance | 7.0107 | 0.00% | 8.5595 | 22.09% | 13.7677 | 96.38% | 7.4723 | 6.58% |
| 3D Maximum Distance | 13.9553 | 0.00% | 26.9987 | 93.47% | 15.1785 | 8.77% | 16.8121 | 20.47% |
| Geographical Distance | 7.7222 | 0.00% | 14.5277 | 88.13% | 16.3916 | 112.27% | 7.9402 | 2.82% |
| Obstructed Distance | 6.5329 | 0.00% | 7.3029 | 11.79% | 14.0040 | 114.36% | 6.9883 | 6.97% |

Table 2: Performance of the MatNet (Kwon et al., 2021) models under different training data configurations on instances with multiple distance metrics. MatNet-SMT, MatNet-Random, and MatNet-MMT refer to the MatNet (Kwon et al., 2021) models trained under the SMT, Random, and MMT configurations, respectively. Average objective value (Obj.) indicates the average solution length. Gap measures the gap of objective value to Concorde (Applegate et al., 2006). The Obstructed Distance refers to the obstructed 2D Euclidean distance.

The experimental results are summarized in Table 2. MatNet-Random performs poorly on most test instances, indicating that training on a single random metric fails to learn a generalizable solving strategy. In contrast, MatNet-MMT achieves balanced generalization. It yields significant improvements over MatNet-SMT. Notably, MatNet-MMT not only performs well on test instances with the same distance metrics as the training instances (e.g., the 2D Euclidean, 3D Euclidean, and geographical distance), but also shows significant improvements on test instances with unseen distance metrics (e.g., the 2D Manhattan, 3D Maximum, and obstructed 2D Euclidean distance). These results suggest that using the MMT training configuration enables the model to effectively learn a general strategy for solving instances under varying distance metrics, leading to better distance metric generalization.

## 5 EXPERIMENTS ON REAL-WORLD VRP INSTANCES

In this section, we evaluate the impact of incorporating distance metric generalization methods on solving real-world problems. Real-world VRPs exhibit diversity in terms of problem scale, node distribution, and distance metric. We combine methods that enhance distance metric generalization with existing advances in generalization across problem scale and node distribution, and conduct a systematic evaluation on real-world VRP instances.

## 5.1 EXPERIMENTAL SETUPS

**Training Data Configurations**    We adopt three distinct training data configurations.

- Single-Metric Training (SMT): This configuration is identical to the Single-Metric Training configuration introduced in Section 4, where the model is trained solely on instances with a single distance metric and a single node distribution.

- Multi-Distribution Training (MDT): Under this configuration, the model is trained solely on instances with the 2D Euclidean distance. In addition, we consider three different node distributions: 2D-uniform, 2D-cluster, and 2D-explosion. There are three types of training instances in total. During training, only a type of training instances is used per epoch. The three types of training instances are alternated sequentially across epochs.

- Multi-Distribution-Multi-Metric Training (MDMMT): Under this configuration, training instances incorporate three different distance metrics: 2D Euclidean, 3D Euclidean, and geographical distance. For each metric, we consider three different node distributions, yielding nine types of training instances in total: 2D-uniform, 2D-cluster, and 2D-explosion, 3D-uniform, 3D-cluster, 3D-explosion, GEO-global, GEO-cluster, and GEO-local. The three node distributions implemented under the 2D Euclidean distance metric are identical to those used in the Multi-Distribution training configuration. During training, only a type of training instances is used per epoch. The nine types of training instances are alternated sequentially across epochs.

All training configurations are conducted on TSP100 instances. The total number of training instances of each configuration keeps constant. All instances are represented as distance matrices and distance matrices are normalized. More implementation details of training instances are provided in the Appendix B.

**Test Dataset**    We evaluated models on a diverse set of publicly available real-world VRP instances. These instances incorporate multiple distance metrics, including 2D Euclidean, 3D Euclidean, geographical, EXPLICIT, Google Maps (Google., 2025), and Open Source Routing Machine (OSRM) (Luxen & Vetter, 2011) distance. The 2D Euclidean, 3D Euclidean and geographical distance are introduced in Section 3.1. Instances with the EXPLICIT distance are from TSPLIB (Reinelt, 1991) and their distance matrices are provided explicitly. Both Google Maps and OSRM provide real-world road network distances. Distances between nodes are obtained by supplying pairs of node coordinates, represented by latitude and longitude, to either Google Maps or OSRM. The problem scales of all instances range from 100 to 10,000 nodes. Detailed results and the sources of these real-world VRP instances are provided in the Appendix C.

**Inference Strategy**    To handle the varying problem scales of real-world instances during inference, we adopted a strategy similar to those proposed by Luo et al. (2024) and Luo et al. (2023). During inference, an initial solution is generated by Random Insertion, which is a heuristic. This solution is then iteratively improved for 100 iterations. In each iteration, a partial sequence of the current solution is destroyed and re-generated by the NCO model. Each partial sequence contains 100 nodes. The newly generated partial sequence replaces the original one if it yields a shorter length.

The training hyperparameters, model hyperparameters, and hardware remain identical to those described in Section 3.2.

## 5.2 RESULTS AND ANALYSIS

The experimental results on real-world VRP instances are summarized in Table 3. MatNet-MDT achieves an improvement of 1.73% in the average gap over MatNet-SMT. However, the improvement is primarily concentrated on a few distance metrics (e.g., 2D Euclidean distance), with little or no improvement on other metrics (e.g., 3D Euclidean distance, EXPLICIT distance, and OSRM distance). These results demonstrate that MatNet-MDT still lacks the generalization ability across varying distance metrics. These findings indicate that generalization across node distributions and across distance metrics are distinct dimensions. Enhancing the generalization across node distributions does not lead to better generalization across distance metrics.

| Distance metric (#) | Random Insertion | MatNet-SMT | MatNet-MDT | MatNet-MDMMT |
| --- | --- | --- | --- | --- |
| | Avg.gap | Avg.gap | Avg.gap | Avg.gap |
| 2D Euclidean Distance (61) | 12.68% | 8.46% | 6.73% | 5.43% |
| 3D Euclidean Distance (2) | 14.75% | 14.75% | 14.75% | 10.45% |
| Geographical Distance (6) | 10.14% | 8.89% | 5.26% | 3.76% |
| EXPLICIT Distance (5) | 7.71% | 7.71% | 7.71% | 5.10% |
| Google Maps Distance (2) | 14.48% | 13.07% | 12.02% | 9.87% |
| OSRM Distance (2) | 12.09% | 11.95% | 11.90% | 10.07% |
| All Instances (78) | 12.25% | 8.82% | 7.15% | 5.64% |

Table 3: Performance of the MatNet (Kwon et al., 2021) models under different training data config-
urations on real-world VRP instances. MatNet-SMT, MatNet-MDT, and MatNet-MDMMT respec-
tively refer to the MatNet (Kwon et al., 2021) models trained under the SMT, MDT, and MDMMT
configurations. Avg.gap measures the average gap of solution length to the best known solution
length. Distance metrics (#) indicates that there are # real-world VRP instances with this distance
metric.

In contrast, MatNet-MDMMT, which is trained on instances with both diverse distance metrics and
node distributions, yields significant performance improvements across all distance metrics com-
pared to MatNet-SMT and MatNet-MDT. This result underscores that the Multi-Distribution-Multi-
Metric training configuration enables the model to adapt more effectively to diverse real-world sce-
narios and achieve stronger robustness.

Therefore, to enable robust real-world application of NCO models, it is critical to consider not only
generalization across problem scales and node distributions, but also the distance metric generaliza-
tion.

## 6  CONCLUSION AND FUTURE WORK

**Conclusion**    The distance metric is an underexplored aspect of generalization in current NCO
methods. In this work, we have introduced a benchmarking framework that encompasses eight
distinct distance metrics and analyzed the distance metric generalization of NCO models. The first
experiment revisits the common training configuration employed by existing NCO models (i.e.,
training on instances with the 2D Euclidean distance) and demonstrates that models trained under
this configuration suffer a notable performance decline when deployed to instances with other dis-
tance metrics. Subsequently, to improve the distance metric generalization of NCO models, we
investigate the impact of various training configurations on the distance metric. The results demon-
strate that training with the Multi-Metric training configuration is most effective in enhancing the
model's generalization. Last but not least, we integrate the Multi-Metric training configuration with
existing advances for generalization across problem scales and node distributions. The experimental
results on real-world VRP instances show that such an integration can further enhance model per-
formance. These findings suggest that improving the distance metric generalization is a critical step
toward applying NCO methods in real-world scenarios. The introduced benchmarking framework
offers a evaluation platform to support future research on the distance metric generalization.

**Limitation and Future Work**    As a pilot study, this paper proposes a simple method called the
Multi-Metric training and demonstrates its effectiveness for improving model generalization across
distance metrics. In future work, we will develop more sophisticated methods that consider not
only training data but also model architectures. Besides, the benchmarking framework in this paper
supports eight distance metrics, and we plan to extend it to include more metrics.

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

# A  PROBLEM DEFINITION OF TSP

The traveling salesman problem (TSP) is one of the most representative VRPs. Let $\{C = c_{j,k}, j = 1, \ldots, n, k = 1 \ldots, n\}$ be the $n \times n$ distance matrix of a TSP instance with a problem scale of $n$, where $c_{j,k}$ denotes the distance between nodes $j$ and $k$. The distance is computed by the corresponding distance metric. The goal of TSP is to minimize the following equation:

$$f(\boldsymbol{x}) = \sum_{t=1}^{n-1} c_{x_t, x_{t+1}} + c_{x_n, x_1}, \tag{8}$$

where $\boldsymbol{x}$ denotes a feasible solution of TSP, which starts from an arbitrary node, visits each node exactly once, and returns to the starting node in the end.

# B  NODE DISTRIBUTIONS FOR INSTANCES WITH DIFFERENT DISTANCE METRICS

The MatNet-MDMMT is trained on instances that incorporate three different distance metrics: 2D Euclidean, 3D Euclidean, and geographical distance. For each metric, the instances cover three distinct node distributions. These node distributions are illustrated in Figure 2.

For the 2D Euclidean distance, we adopt three distinct distributions: 2D-uniform, 2D-cluster, and 2D-explosion. The specific implementations follow Fang et al. (2024).

**2D-uniform**  Node coordinates are uniformly sampled within a unit square.

**2D-cluster**  First, $c$ cluster centers are uniformly sampled in the range $[0, L]^2$, and $n$ nodes are uniformly sampled within $[0, 1]^2$. Each node is randomly assigned to one of the clusters, and its coordinates then add the coordinates of its assigned cluster center. Finally, all coordinates are normalized back to the $[0, 1]^2$. In our experiments on TSP100 (i.e., $n = 100$), we set $L = 10$ and $c = 3$.

**2D-explosion**  An explosion center and node coordinates are initially uniformly sampled in $[0, 1]^2$. A radius $r$ uniformly sampled in $[r_{min}, r_{max}]$. All nodes located inside the disk centered at the explosion center with radius $r$ are displaced outward according to an exponential distribution with rate $\lambda$. The parameters are set as $r_{min} = 0.1$, $r_m ax = 0.5$, and $\lambda = 10$.

For the 3D Euclidean distance, we adopt three distinct distributions: 3D-uniform, 3D-cluster, and 3D-explosion. These are the direct 3D extensions of the node distribution in 2D space.

**3D-uniform**  Node coordinates are uniformly sampled within a unit cube.

**3D-cluster**  First, $c$ cluster centers are uniformly sampled in the range $[0, L]^3$, and $n$ nodes are uniformly sampled within $[0, 1]^3$. Each node is randomly assigned to one of the clusters, and its coordinates then add the coordinates of its assigned cluster center. Finally, all coordinates are normalized to the $[0, 1]^3$. In our experiments on TSP100 (i.e., $n = 100$), we set $L = 10$ and $c = 3$.

**3D-explosion**  An explosion center and node coordinates are initially uniformly sampled in $[0, 1]^3$. A radius $r$ uniformly sampled in $[r_{min}, r_{max}]$. All nodes located inside the sphere centered at the explosion center with radius $r$ are displaced outward according to an exponential distribution with rate $\lambda$. The parameters are set as $r_{min} = 0.2$, $r_{max} = 0.8$, and $\lambda = 10$.

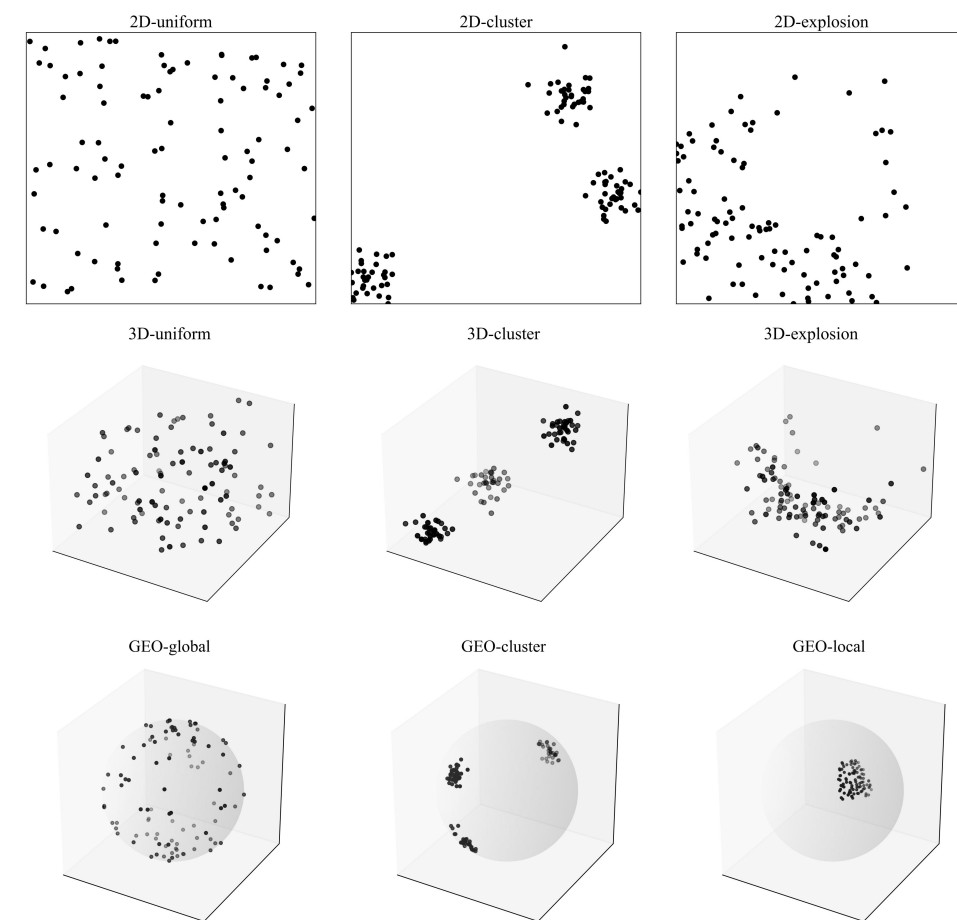

Figure 2: Visualization of TSP100 instances with various distributions. Each sub-figure illustrates an instance that follows its specified distribution.

For the geographical distance, we adopt three distinct distributions: GEO-global, GEO-cluster, and GEO-local.

**GEO-global**  Node coordinates are uniformly sampled within the feasible ranges (i.e., longitude: $[-180, 180]$, latitude: $[-90, 90]$).

**GEO-cluster**  $c$ cluster centers are uniformly sampled in the latitude and longitude range $[-90, 90]$ and $[-180, 180]$, respectively. $n$ nodes are randomly assigned to one of these clusters, with each node's initial coordinates set to its assigned cluster center. Then, gaussian noise with mean 0 and standard deviation $\sigma$ is added to each node's coordinates. In our experiments on TSP100 (i.e., $n = 100$), we set $c = 3$ and $\sigma = 6$.

**GEO-local**  A center point $(\lambda, \phi)$ of a local region is first uniformly sampled from the feasible range of coordinates (i.e., longitude: $[-180, 180]$, latitude: $[-90, 90]$). hen, node coordinates are uniformly sampled within this local region, where the longitude is in the range $[\lambda - \delta, \lambda + \delta]$ and the latitude is in the range $[\phi - \delta, \phi + \delta]$. In our experiments, we set $\delta = 2$.

## C  DETAILED RESULTS AND SOURCES OF REAL-WORLD TSP INSTANCES

The detailed results and sources on real-world TSP instances datasets are presented in Tables 4 and 5.

| Distance Metric | Instance | Source | Random Insertion | MatNet-SMT | MatNet-MDT | MatNet-MDMMT |
|---|---|---|---|---|---|---|
| | eil101 | TSPLIB | 8.59% | 2.07% | 1.43% | 1.43% |
| | lin105 | TSPLIB | 14.85% | 5.24% | 2.45% | 1.09% |
| | pr107 | TSPLIB | 1.70% | 1.70% | 1.70% | 1.65% |
| | pr124 | TSPLIB | 13.74% | 2.40% | 1.15% | 0.26% |
| | bier127 | TSPLIB | 8.40% | 7.77% | 1.98% | 1.25% |
| | ch130 | TSPLIB | 12.55% | 1.10% | 1.00% | 0.62% |
| | pr136 | TSPLIB | 9.59% | 1.91% | 1.18% | 1.04% |
| | pr144 | TSPLIB | 10.68% | 5.64% | 4.17% | 2.81% |
| | ch150 | TSPLIB | 7.75% | 0.77% | 1.75% | 1.69% |
| | kroA150 | TSPLIB | 7.13% | 2.12% | 1.82% | 1.25% |
| | kroB150 | TSPLIB | 6.41% | 1.81% | 0.77% | 0.51% |
| | pr152 | TSPLIB | 4.86% | 4.80% | 3.56% | 3.67% |
| | u159 | TSPLIB | 12.13% | 2.09% | 0.76% | 0.00% |
| | rat195 | TSPLIB | 12.79% | 2.71% | 1.21% | 1.03% |
| | d198 | TSPLIB | 9.60% | 9.33% | 3.15% | 2.47% |
| | kroA200 | TSPLIB | 12.48% | 3.38% | 1.41% | 0.91% |
| | kroB200 | TSPLIB | 12.18% | 2.08% | 3.97% | 2.99% |
| | ts225 | TSPLIB | 11.74% | 1.81% | 1.12% | 1.16% |
| | tsp225 | TSPLIB | 8.86% | 4.55% | 2.81% | 1.56% |
| | pr226 | TSPLIB | 5.06% | 5.06% | 4.61% | 3.98% |
| | gil262 | TSPLIB | 12.83% | 5.93% | 4.96% | 3.41% |
| | pr264 | TSPLIB | 8.16% | 8.16% | 3.99% | 0.88% |
| | a280 | TSPLIB | 19.43% | 8.41% | 4.69% | 3.57% |
| | pr299 | TSPLIB | 14.08% | 7.22% | 3.70% | 2.73% |
| | lin318 | TSPLIB | 8.11% | 6.71% | 3.87% | 3.08% |
| | linhp318 | TSPLIB | 11.88% | 8.67% | 7.11% | 4.78% |
| | rd400 | TSPLIB | 12.50% | 8.23% | 4.94% | 3.89% |
| | fl417 | TSPLIB | 12.55% | 12.55% | 12.55% | 12.40% |
| | pr439 | TSPLIB | 9.76% | 8.38% | 5.48% | 3.98% |
| | pcb442 | TSPLIB | 13.82% | 6.12% | 6.47% | 5.15% |
| 2D Euclidean Distance | d493 | TSPLIB | 9.34% | 6.82% | 5.68% | 4.20% |
| | u574 | TSPLIB | 10.40% | 6.61% | 6.07% | 5.41% |
| | rat575 | TSPLIB | 12.11% | 8.71% | 7.71% | 4.12% |
| | p654 | TSPLIB | 6.60% | 6.60% | 6.23% | 6.49% |
| | d657 | TSPLIB | 12.89% | 7.05% | 5.88% | 4.22% |
| | u724 | TSPLIB | 12.05% | 8.06% | 7.12% | 4.14% |
| | rat783 | TSPLIB | 14.34% | 10.94% | 8.52% | 6.19% |
| | pr1002 | TSPLIB | 13.64% | 10.35% | 8.94% | 6.22% |
| | u1060 | TSPLIB | 11.98% | 9.78% | 6.93% | 5.40% |
| | vm1084 | TSPLIB | 13.33% | 9.93% | 8.50% | 6.82% |
| | pcb1173 | TSPLIB | 16.09% | 11.83% | 9.29% | 8.50% |
| | d1291 | TSPLIB | 17.62% | 16.15% | 12.56% | 8.96% |
| | rl1304 | TSPLIB | 19.79% | 17.04% | 13.48% | 11.92% |
| | rl1323 | TSPLIB | 19.64% | 13.09% | 12.30% | 9.17% |
| | nrw1379 | TSPLIB | 11.61% | 8.42% | 7.01% | 5.33% |
| | fl1400 | TSPLIB | 6.71% | 6.50% | 6.22% | 5.93% |
| | u1432 | TSPLIB | 12.31% | 8.73% | 7.49% | 5.53% |
| | fl1577 | TSPLIB | 14.82% | 12.71% | 9.41% | 8.60% |
| | d1655 | TSPLIB | 18.51% | 16.20% | 12.81% | 9.90% |
| | vm1748 | TSPLIB | 13.11% | 11.83% | 8.25% | 7.46% |
| | u1817 | TSPLIB | 18.43% | 15.23% | 11.55% | 9.10% |
| | rl1889 | TSPLIB | 18.40% | 16.64% | 14.87% | 12.88% |
| | d2103 | TSPLIB | 23.08% | 18.67% | 15.11% | 13.28% |
| | u2152 | TSPLIB | 20.15% | 13.17% | 11.36% | 9.76% |
| | u2319 | TSPLIB | 6.94% | 4.53% | 3.72% | 2.65% |
| | pr2392 | TSPLIB | 16.52% | 13.31% | 12.63% | 10.52% |
| | pcb3038 | TSPLIB | 16.42% | 12.59% | 11.87% | 9.34% |
| | fl3795 | TSPLIB | 14.52% | 13.51% | 11.30% | 10.61% |
| | fnl4461 | TSPLIB | 12.65% | 10.96% | 10.01% | 8.14% |
| | rl5915 | TSPLIB | 23.59% | 21.86% | 19.30% | 18.27% |
| | rl5934 | TSPLIB | 21.50% | 19.63% | 18.73% | 16.67% |

Table 4: Gap to the best known solution (BKS) on real-world TSP instances.

## D  LLM USAGE

In the preparation of this manuscript, the authors employed large language models (LLMs) as a tool to assist in the writing and refinement process. The primary role of the LLMs was to enhance the readability, fluency, and grammatical accuracy of the text. Specifically, they were used to rephrase complex ideas, suggest more concise formulations, and ensure that the language adhered to academic standards.

| Distance Metric | Instance | Source | Random Insertion | MatNet-SMT | MatNet-MDT | MatNet-MDMMT |
|---|---|---|---|---|---|---|
| Geographical Distance | gr137 | TSPLIB | 6.40% | 6.40% | 0.80% | 1.08% |
| | gr202 | TSPLIB | 9.30% | 7.40% | 4.85% | 2.81% |
| | gr229 | TSPLIB | 7.85% | 6.77% | 3.32% | 2.36% |
| | gr431 | TSPLIB | 13.18% | 10.01% | 5.12% | 3.81% |
| | ali535 | TSPLIB | 11.26% | 10.43% | 6.93% | 3.95% |
| | gr666 | TSPLIB | 12.87% | 12.33% | 10.57% | 8.57% |
| EXPLICIT Distance | gr120 | TSPLIB | 11.61% | 11.61% | 11.61% | 6.60% |
| | si175 | TSPLIB | 2.88% | 2.88% | 2.88% | 2.37% |
| | si535 | TSPLIB | 2.06% | 2.06% | 2.06% | 2.06% |
| | pa561 | TSPLIB | 16.50% | 16.50% | 16.50% | 10.03% |
| | si1032 | TSPLIB | 5.47% | 5.47% | 5.47% | 4.45% |
| 3D Euclidean Distance | star1k | Website[1] | 14.94% | 14.94% | 14.94% | 7.94% |
| | star10k | Website[2] | 14.57% | 14.57% | 14.57% | 12.97% |
| Google Maps Distance | college647 | Website[3] | 11.92% | 10.26% | 9.17% | 7.61% |
| | usa3100 | Website[4] | 17.05% | 15.88% | 14.87% | 12.13% |
| OSRM Distance | korea383 | Website[5] | 11.12% | 10.84% | 10.84% | 7.43% |
| | dj2141 | Website[5] | 13.06% | 13.06% | 12.97% | 12.70% |

Table 5: Gap to the best known solution (BKS) on real-world TSP instances.

# E    LICENSES

The licenses for the codes, data instances, and datasets used in this work are listed in Table 6.

| Resources | Type | Link | License |
|---|---|---|---|
| Concorde | Code | https://github.com/jvkersch/pyconcorde | BSD 3-Clause License |
| MatNet | Code | https://github.com/henry-yeh/GLOP | MIT License |
| TSPLIB | Dataset | http://comopt.ifi.uni-heidelberg.de/software/TSPLIB95/ | Available for any non-commercial use |
| star1k | Data Instance | https://www.math.uwaterloo.ca/tsp/star/star1k.html | Available for any non-commercial use |
| star10k | Data Instance | https://www.math.uwaterloo.ca/tsp/star/star10k.html | Available for any non-commercial use |
| korea383 | Data Instance | https://www.math.uwaterloo.ca/tsp/korea/more.html | Available for any non-commercial use |
| dj2141 | Data Instance | https://www.math.uwaterloo.ca/tsp/korea/more.html | Available for any non-commercial use |
| college647 | Data Instance | https://www.math.uwaterloo.ca/tsp/college/index.html | Available for any non-commercial use |
| usa3100 | Data Instance | https://www.math.uwaterloo.ca/tsp/county/index.html | Available for any non-commercial use |

Table 6: A summary of licenses.

---

[1] https://www.math.uwaterloo.ca/tsp/star/star1k.html
[2] https://www.math.uwaterloo.ca/tsp/star/star10k.html
[3] https://www.math.uwaterloo.ca/tsp/college/index.html
[4] https://www.math.uwaterloo.ca/tsp/county/index.html
[5] https://www.math.uwaterloo.ca/tsp/korea/more.html

