# OpenReview forum: "Rethinking Distance Metric Generalization in Neural Combinatorial Optimization for Vehicle Routing Problems"
_ICLR.cc/2026/Conference — Submitted to ICLR 2026_

### Official Review · Reviewer_RE4j · 2025-10-20

**Soundness:** 1
**Presentation:** 2
**Contribution:** 1
**Rating:** 2
**Confidence:** 5

**Summary:**

This manuscript proposes a Multi-Metric training strategy.

**Strengths:**

This manuscript is well-structured.

**Weaknesses:**

**W1 Limited innovation:** Although the authors proposed a Multi-Metric training strategy, it is relatively simple. Moreover, joint training approaches of this type have already been widely adopted in other neural combinatorial optimization studies [1-3].

**W2 Insufficient experiments:**  The experimental evaluation is limited to applying MDMMT on MatNet, which constrains the evidence for the method’s generality.



[1] Multi-Task Learning for Routing Problem with Cross-Problem Zero-Shot Generalization. KDD, 2024.

[2] CaDA: Cross-Problem Routing Solver with Constraint-Aware Dual-Attention. ICML, 2025.

[3] RouteFinder: Towards Foundation Models for Vehicle Routing Problems. TMLR, 2025.

**Questions:**

See Weaknesses.

---

> ### Author Response · Authors · 2025-11-25
> **Response to Reviewer RE4j**
>
> Thank you very much for your time and effort in reviewing our work. We are glad to know you find our manuscript is well-structured.
>
> We address your concerns as follows.
>
> **Weakness 1. Limited innovation:**
>
> Please check the response of the Weakness 1 in the General Response.
>
> **Weakness 2. Insufficient experiments:**
>
> Please check the response of the Weakness 2 in the General Response.

---

> > ### Comment · Reviewer_RE4j · 2025-11-26
> >
> > Thank you for your responses.

---

### Official Review · Reviewer_Y6ZH · 2025-10-20

**Soundness:** 3
**Presentation:** 2
**Contribution:** 1
**Rating:** 2
**Confidence:** 4

**Summary:**

This paper investigates distance metric generalization in vehicle routing problems (VRPs). It introduces a benchmarking framework covering eight distinct distance metrics to evaluate the generalization ability of neural combinatorial optimization (NCO) models. Furthermore, it proposes a simple multi-task training approach as a solution for this setting. However, this paper still has several notable issues, and in its current form, the contributions may not yet be sufficient for acceptance at a top-tier ML conference.

**Strengths:**

* This paper investigates distance metric generalization, a key challenge that limits the applicability of NCO approaches in real-world scenarios. The proposed generalization setting deserves more attention from the NCO community.
* The writing of the abstract and introduction is clear and well-structured.

**Weaknesses:**

* This paper studies distance metric generalization in VRPs. However, a recent work [1] has already explored this setting. The authors should discuss and highlight the differences and contributions beyond prior work.
* Section 3 presents a benchmark for distance metric generalization, but it only includes one model (MatNet) and one training setting (2D Euclidean). The benchmark would be more comprehensive if it included additional approaches and settings, such as those mentioned in Section 2 `Distance-Matrix-Based`.
* It is rather straightforward that multi-metric training can enhance model generalization across distance metrics. The proposed method therefore offers limited technical novelty, and deeper insights or analyses are needed.
* The paper title refers to VRPs, yet the experiments are limited to TSP. Additional problem variants (e.g., CVRP) and stronger baselines should be included to validate generality.
* Beyond the abstract and introduction, the writing of the remaining sections requires improvement for clarity and flow.

[1] Lifelong Learner: Discovering Versatile Neural Solvers for Vehicle Routing Problems.

**Questions:**

* Why are only three distance metrics considered for multi-metric training (MMT) in Section 4?
* Why not considered asymmetric distance?
* Although the distance-matrix-based input format is more general, it incurs an $O(N^2)$ space complexity. What is the computational overhead when training or testing on large-scale instances as reported in Table 4?

---

> ### Author Response · Authors · 2025-11-25
> **Response to Reviewer Y6ZH**
>
> Thank you very much for your time and effort in reviewing our work. We are glad to know that you find that the proposed generalization setting deserves more attention from the NCO community and the writing of the abstract and introduction is clear and well-structured.
>
> We address your concerns point-by-point as follows:
>
>
>
> **Weakness 1. Difference with a recent work:**
>
> Per ICLR policy (see the last question of reviewer FAQ), we are not required to compare our work with papers published after July 24, 2025, or solely on arXiv. As [1] was released on arXiv on August 8, 2025, we are excused from mentioning it
>
> Despite this, we explain how our contributions are different from [1]:
>
> - Our work considers models that take distance matrices as input. In contrast, [1] considers models that use node coordinates as input. As mentioned in our introduction, the coordinate-based input format inherently couples the model with some specific distance metrics, limiting its applicability to other distance metrics. Consequently, coordinate-based models inherently exhibit limitations in investigating generalization across diverse distance metrics.
>
> - Our work evaluates models on instances with eight different distance metrics. In contrast, [1] evaluates models only on instances with three distance metrics (i.e., 2D Euclidean, 2D Manhattan, and 2D Chebyshev (i.e., 2D Maximum)). We considered a broader range of distance metrics than [1].
>
>
>
> **Weakness 2. Additional approaches and settings:**
>
> Please check the response of the Weakness 2 in the General Response.
>
>
>
> **Weakness 3. Method is straightforward:**
>
> Please check the response of the Weakness 1 in the General Response.
>
>
>
> **Weakness 4. CVRP experiments:**
>
> We add experiments on CVRP-100 and a stronger baseline (i.e., GOAL [2]) to assess generality beyond TSP.
> The metrics chosen for SMT and MMT are the same as in the experiments on TSP.
> As shown in the following table, simply training model with diverse distance metrics can effective improve model generalization across different metrics.
>
>
>
> |                       | GOAL-CVRP-SMT | GOAL-CVRP-MMT |
> | --------------------- | ------------- | ------------- |
> | 2D Euclidean Distance | 6.95%         | 6.81%         |
> | 3D Euclidean Distance | 5.31%         | 4.74%         |
> | 2D Manhattan Distance | 7.69%         | 7.15%         |
> | 3D Manhattan Distance | 5.79%         | 4.89%         |
> | 2D Maximum Distance   | 7.69%         | 7.12%         |
> | 3D Maximum Distance   | 6.04%         | 5.02%         |
> | Geographical Distance | 7.32%         | 6.49%         |
>
>
>
> **Question 1. Why are only three distance metrics considered for multi-metric training (MMT) in Section 4?**
>
> To validate the effectiveness of the MMT and rigorously assess generalization rather than overfitting the metrics present in training, we partition the available metrics into a training set and a test set for experiments. Thus, three distance metrics considered for MMT and five distance metrics are used to rigorously evaluate the generalization performance on instances with unseen metrics.
>
>
>
> **Question 2. Why not considered asymmetric distance?**
>
> Thanks for your comments. Asymmetry is not a distance metric. Our work aims to assess the impact of distance metrics on model generalization. Introducing asymmetric distances confounds the influence of the distance metric and symmetry, making it difficult to attribute performance changes to either factor.
>
>
>
> **Question 3. Computational overhead on large-scale instances:**
>
> Regarding the experiments in Table 4, the models are trained exclusively on small-scale instances (i.e., TSP100).
>
> During inference on large-scale instances, we employ a destroy-and-regenerate inference strategy. An initial solution is constructed via Random Insertion and then iteratively refined. In each iteration, a segment of the current solution comprising exactly 100 nodes is destroyed. Then, the model regenerates this segment. Since it processes only the 100 nodes within this segment at a time, the input to the model in every iteration is a distance matrix of fixed dimensions 100×100.
>
> Therefore, regardless of the instance scale, the space complexity remains constant (i.e., O(1)).
>
>
>
>
>
> **References**
>
> [1] Feng S, Lin Z, Zhou J, et al. Lifelong Learner: Discovering Versatile Neural Solvers for Vehicle Routing Problems[J]. arXiv preprint arXiv:2508.11679, 2025.
>
> [2] Drakulic D, Michel S, Andreoli J M. GOAL: A generalist combinatorial optimization agent learner[J]. arXiv preprint arXiv:2406.15079, 2024.

---

> > ### Comment · Reviewer_Y6ZH · 2025-11-26
> >
> > Thanks for your rebuttal. I will take it into consideration for my final decision.

---

### Official Review · Reviewer_BwPX · 2025-10-22

**Soundness:** 2
**Presentation:** 3
**Contribution:** 2
**Rating:** 4
**Confidence:** 5

**Summary:**

This paper addresses the issue of limited distance metric generalization in neural combinatorial optimization (NCO) models applied to the Vehicle Routing Problem (VRP). It highlights the dependency of current NCO methods on a single distance metric (e.g., 2D Euclidean distance), which restricts their ability to generalize to different distance metrics encountered in real-world scenarios (e.g., 3D Euclidean, geographical distances). The authors propose a benchmarking framework that supports eight different distance metrics to evaluate NCO models. They introduce a "Multi-Metric Training (MMT)" strategy, which significantly improves generalization across various distance metrics. Experimental results on VRPLIB instances show the effectiveness of MMT.

**Strengths:**

1. This work is relevant for the practical deployment of NCO models in real-world VRP scenarios, where diverse distance metrics are common.
2. The methodology is clearly explained, and the paper is easy to follow.

**Weaknesses:**

1. The MMT strategy is based on only four distance metrics. The authors do not explain why these particular metrics were selected and how well they represent the diversity of real-world metrics. Additional distance metrics or a more diverse selection should be included.
2. The analysis of *why* different distance metrics affect model performance feels a bit superficial. The paper shows that single-metric training leads to poor generalization but doesn’t really explore what properties of distance metrics (e.g., symmetry, triangle inequality strength) cause this. Some theoretical or interpretive discussion would make the contribution stronger.
3. The MMT approach is quite straightforward — essentially an application of standard multi-task training — without much adaptation to the specific characteristics of distance metrics.
4. The author should consider other state-of-the-art distance-matrix-based models as baselines instead of MatNet (2021), as discussed in Related Work.

**Questions:**

1. The paper defines a “Maximum Distance” metric, but it’s unclear where such a metric would appear in real-world routing scenarios. Could the authors give an example of its practical use?
2. How were the metrics chosen for MMT training? Were other metrics (e.g., Manhattan, Maximum, obstructed) tested and found less effective?

---

> ### Author Response · Authors · 2025-11-25
> **Response to Reviewer BwPX**
>
> Thank you very much for your time and effort in reviewing our work. We are glad to know that you find that the paper is relevant for the practical deployment of NCO models in real-world VRP scenarios and our paper is easy to follow.
>
> We address your concerns as follows.
>
> **Weakness 1. Selection of MMT training set:**
>
> We thank the reviewer for raising this insightful concern regarding the selection of MMT training set.
>
> The primary objective of the MMT experiments is not to identify the optimal selection of MMT training set but to illustrate the limitations of Single-Metric Training (SMT) as the prevailing training configuration. We aim to demonstrate that simply employing multiple distance metrics for joint training (MMT) yields better generalization than Single-Metric Training (SMT).
>
> To further support our argument, we conduct additional experiments. We trained the model using the combination (2D Manhattan, 2D Maximum, Obstructed), denoted as MatNet-MMT (Man, Max, Obs). Results are reported in the following tables.
>
> |                                  | MatNet-SMT | MatNet-MMT (Man, Max, Obs) |
> | -------------------------------- | ---------- | -------------------------- |
> | 2D Euclidean Distance            | 2.03%      | 1.09%                      |
> | 3D Euclidean Distance            | 67.15%     | 47.83%                     |
> | 2D Manhattan Distance            | 14.74%     | 1.74%                      |
> | 3D Manhattan Distance            | 67.52%     | 44.18%                     |
> | 2D Maximum Distance              | 22.09%     | 1.45%                      |
> | 3D Maximum Distance              | 93.47%     | 58.91%                     |
> | Geographical Distance            | 88.13%     | 40.17%                     |
> | Obstructed 2D Euclidean Distance | 12.57%     | 1.59%                      |
>
> These results indicate that MatNet-MMT (Man, Max, Obs) outperforms MatNet-SMT across all evaluated distance metrics. These results further strengthen our claim that simply incorporating multiple distance metrics (MMT) can yield better generalization than SMT.
>
> **Weakness 2. Theoretical or interpretive discussion:**
>
> Thanks for your insightful comments. We acknowledge that analyzing the intrinsic properties of distance metrics is important for understanding their impact on model generalization. In this work, we aim to provide an initial exploration of the impact of different distance metrics, focusing primarily on empirical comparisons. In the future, we plan to conduct a deeper investigation with theoretical or interpretive discussions to better understand why different distance metrics affect model performance.
>
> Notably, all the distance metrics used in our experiments satisfy the properties you mentioned (i.e., symmetry and triangle inequality). Therefore, these properties are unlikely to be the primary cause of the observed differences in generalization performance.
>
> **Weakness 3. Approach is quite straightforward:**
>
> Please check the response of the Weakness 1 in the General Response.
>
> **Weakness 4. Consider other state-of-the-art distance-matrix-based models as baselines:**
>
> Please check the response of the Weakness 2 in the General Response.
>
> **Question 1. Real-world VRP applications for the Maximum distance metrics:**
>
> Please check the response of the Question 1 in the General Response.

---

### Official Review · Reviewer_QqGB · 2025-11-06

**Soundness:** 2
**Presentation:** 2
**Contribution:** 2
**Rating:** 2
**Confidence:** 4

**Summary:**

This paper presents an empirical study on the generalization of Neural Combinatorial Optimization (NCO) models across different distance metrics for Vehicle Routing Problems. The authors first demonstrate that models trained on a single metric (e.g., 2D Euclidean) fail to generalize to unseen metrics, as shown in Table 1. To address this, the paper introduces a new benchmark framework encompassing eight distinct distance metrics. The authors also propose a method, "Multi-Metric Training" (MMT), which involves jointly training the model on instances from multiple different metrics. The final proposed model, "MDMMT," combines this approach with existing techniques for distribution generalization to achieve improved performance on real-world VRP instances.

**Strengths:**

This work addresses an important and under-explored aspect of generalization in NCO models. The paper's primary strength is its clear empirical demonstration that standard models overfit to a single distance metric, resulting in a significant performance collapse on unseen metrics (Table 1). The proposed benchmarking framework is a useful contribution to the community for evaluating this specific dimension of robustness.

**Weaknesses:**

The paper's central weakness is the profound lack of methodological novelty. The proposed "Multi-Metric Training (MMT)"  is presented as a novel method, but it is functionally identical to the standard, well-known data augmentation strategy of training on a more diverse dataset. The paper's own text highlights this contradiction: it frames "Multi-Distribution Training (MDT)"  (training on diverse node distributions) as prior art , while presenting MMT (training on diverse metrics) as its own contribution. This is a distinction without a difference; both are simple data augmentation strategies, and the core "insight" that training on diverse data improves generalization on that data is not a sufficient contribution for a top-tier methods conference.

The motivation for the benchmark itself is underdeveloped. While the authors provide real-world applications for 3D Euclidean and Geographical distances , they fail to provide any practical justification for the inclusion of other key metrics, such as 2D/3D Manhattan or 2D/3D Maximum distances. Without this context, these additions appear arbitrary, undermining the practical significance of the benchmark. This supports the concern that experiments are being conducted for their own sake rather than to address clearly defined, practical needs.

The paper's contribution appears incremental when viewed in the context of recent SOTA "generalist" NCO solvers. Works like UniCO (Pan et al., 2025)  are also leveraging matrix-based inputs  to unify entirely different problem types (e.g., TSP, HCP, SAT) , which is a far more significant generalization challenge than generalizing across different instantiations of a distance matrix.

Furthermore, the "real-world" experiment in Section 5 introduces significant confounding variables. To handle large-scale instances (up to 10k nodes), the paper embeds the NCO model into an iterative 'destroy and re-generate' heuristic framework. The final results in Table 3  show the performance of this combined system, making it impossible to decouple the gains from the MDMMT training strategy itself versus the gains from the powerful iterative search framework. A more rigorous design would have included a head-to-head comparison on medium-scale real-world instances without this heuristic wrapper to isolate the model's true performance.

Finally, the design of the MMT training set is arbitrary and lacks supporting ablation studies. The authors select {2D Euclidean, 3D Euclidean, Geographical} for MMT training  and show it generalizes to unseen metrics like 2D Manhattan. However, the paper provides no analysis as to why this specific combination is effective or if it is optimal. It is unknown if another combination (e.g., {2D Euc, 2D Manhattan, Obstructed}) would also generalize to 3D and Geographical metrics. This lack of analysis makes the specific MMT configuration feel arbitrary and its success on unseen metrics insufficiently explained.

Pan, W(2025). "UniCO: On Unified Combinatorial Optimization via Problem Reduction to Matrix-Encoded General TSP." In The Thirteenth International Conference on Learning Representations (ICLR).

**Questions:**

Can you clarify the methodological novelty of MMT? Given that training on diverse data to improve generalization is a standard technique, and that the paper itself treats the analogous "Multi-Distribution Training" (MDT) as prior art , what is the specific, novel contribution of MMT beyond being a simple data augmentation strategy?

What are the specific, real-world VRP applications for the Manhattan and Maximum distance metrics that motivated their inclusion in the benchmark?

How does your approach to "metric generalization" compare to more ambitious "generalist" solvers like UniCO (Pan et al., 2025) , which aim to generalize across different problem types using a similar matrix-based input format?

---

> ### Author Response · Authors · 2025-11-25
> **Response to Reviewer QqGB [1/2]**
>
> Thank you very much for your time and effort in reviewing our work.
> We are glad to know you find our work addresses an important and under-explored aspect of generalization in NCO models and provides a valuable benchmarking framework for the community to evaluate this dimension of robustness.
> We address your concerns as follows.
>
> **Weakness 1. Lack of methodological novelty:**
>
> Please check the response of the Weakness 1 in the General Response.
>
> **Weakness 2. Incremental contribution:**
>
> Thanks for your comments. We respectfully disagree that our contribution is incremental. Our work addresses a challenge that is orthogonal and complementary to that in UniCO:
>
> - UniCO emphasizes a single model's ability to solve diverse combinatorial problems. In contrast, this work focuses on a model's capacity to generalize across instances with varying distance metrics for the same problem.
>
> - A model capable of solving multiple problem types may still struggle to adapt to changes in the distance metric within a single problem type. As shown in the following table, the pretrained model provided by the author of UniCO, which can solve multiple combinatorial problems (i.e., TSP, HCP, SAT), exhibits limited performance when evaluated on TSP instances with diverse distance metrics. Systematically studying and improving generalization across distance metrics is essential for building powerful and general neural combinatorial optimization solvers.
>
> Our paper systematically investigates the generalization across distance metrics and represents a foundational contribution with practical relevance, rather than an incremental contribution.
>
> Table 1: Performance of the  UniCO-MatPOENet-mix on instances with multiple distance metrics.
>
> | | MatNet-MMT | UniCO-MatPOENet-mix |
> |---|---|---|
> | 2D Euclidean Distance | 2.03% | 7.71% |
> | 3D Euclidean Distance | 2.35% | 7.77% |
> | 2D Manhattan Distance | 6.78% | 8.36% |
> | 3D Manhattan Distance | 9.98% | 12.60% |
> | 2D Maximum Distance   | 6.58% | 8.54% |
> | 3D Maximum Distance   | 20.47% | 8.53% |
> | Obstructed 2D Euclidean Distance | 6.97% | 8.65% |

---

> ### Author Response · Authors · 2025-11-25
> **Response to Reviewer QqGB [2/2]**
>
> **Weakness 3. Confounded performance gains:**
>
> Thank you for your comments. We would like to clarify that our experimental design is rigorous and effectively supports our core claims, without confusing the gains of the MDMMT training strategy with the gains of the iterative search framework.
>
> The performance gains reported in Table 3 are clearly attributed to the training strategy. All compared methods (i.e., SMT, MDT, and MDMMT) are embedded within the same iterative destroy-and-regenerate framework. Thus, the improvements of MDMMT over SMT or MDT can be clearly attributed to the enhanced model capability resulting from MDMMT.
>
> MatNet [1] has limitations in handling arbitrarily large matrices, and its implementation imposes a preset maximum size of 256, which which limits its application to large-scale instances [2]. Thus, we adopt a destroy-and-regenerate heuristic framework, which is effective in scaling models to large-scale instances [3]. Additionally, we evaluate our method without the heuristic wrapper on real-world instances with fewer than 256 nodes. Only three distance metrics (i.e., 2D Euclidean, Geographical, and EXPLICIT) include such instances. Results demonstrate that MDMMT still effectively improves generalization across different distance metrics.
>
> Table 2: Performance of the models without the heuristic wrapper on real-world VRP instances with fewer than 256 nodes.
>
> |                            | MatNet-SMT | MatNet-MDT | MatNet-MDMMT |
> | -------------------------- | ---------- | ---------- | ------------ |
> | 2D Euclidean Distance (20) | 21.10%     | 6.71%      | 8.06%        |
> | Geographical Distance (3)  | 34.22%     | 51.59%     | 16.37%       |
> | EXPLICIT Distance (2)      | 44.14%     | 16.80%     | 10.07%       |
> | All Instances (25)         | 24.28%     | 9.41%      | 8.61%        |
>
> **Weakness 4. The metrics chosen for MMT training:**
>
> We thank the reviewer for raising this insightful concern regarding the metrics chosen for MMT training. The primary objective of the MMT experiments is not to identify an optimal combination of training metrics but to illustrate the limitations of Single-Metric Training (SMT) as the prevailing training configuration. We aim to demonstrate that simply employing multiple distance metrics for joint training (MMT) yields better generalization than Single-Metric Training (SMT).
>
> To further support our argument, we conduct additional experiments. We trained the model using the combination (2D Euclidean, 2D Manhattan, Obstructed), denoted as MatNet-MMT (Euc, Man, Obs). Results are reported in the following tables.
>
> |                                  | MatNet-SMT | MatNet-MMT (Euc, Man, Obs) |
> | -------------------------------- | ---------- | -------------------------- |
> | 2D Euclidean Distance            | 2.03%      | 1.02%                      |
> | 3D Euclidean Distance            | 67.15%     | 32.48%                     |
> | 2D Manhattan Distance            | 14.74%     | 1.98%                      |
> | 3D Manhattan Distance            | 67.52%     | 32.50%                     |
> | 2D Maximum Distance              | 22.09%     | 2.94%                      |
> | 3D Maximum Distance              | 93.47%     | 40.45%                     |
> | Geographical Distance            | 88.13%     | 37.94%                     |
> | Obstructed 2D Euclidean Distance | 12.57%     | 1.76%                      |
>
> These results indicate that MatNet-MMT (Euc, Man, Obs) outperforms MatNet-SMT across all evaluated distance metrics. These results further strengthen our claim that simply incorporating multiple distance metrics (MMT) can yield better generalization than SMT. Our work aims to raise awareness regarding the generalization of distance metrics, a common challenge in real-world applications.
>
>
>
> **Question 1. Real-world VRP applications for the Manhattan and Maximum distance metrics:**
>
> Please check the response of the Question 1 in the General Response.
>
>
>
> **Question 2. Compare to UniCO:**
>
> Please check the response of the Weakness 2.
>
>
>
> **References**
>
> [1] Kwon Y D, Choo J, Yoon I, et al. Matrix encoding networks for neural combinatorial optimization[J]. Advances in Neural Information Processing Systems, 2021, 34: 5138-5149.
>
> [2] Pan W, Xiong H, Ma J, et al. UniCO: On unified combinatorial optimization via problem reduction to matrix-encoded general TSP[C]//The Thirteenth International Conference on Learning Representations. 2025.
>
> [3] Ye H, Wang J, Liang H, et al. Glop: Learning global partition and local construction for solving large-scale routing problems in real-time[C]//Proceedings of the AAAI conference on artificial intelligence. 2024, 38(18): 20284-20292.

---

### Author Response · Authors · 2025-11-25
**General Response [1/2]**

We would like to thank all the reviewers for their efforts in reviewing our paper and providing us with helpful comments. We provide answers to some general weaknesses and questions.

**Weakness 1. Quite straightforward method with limited methodological novelty:**

We would like to clarify the core positioning and contribution of our work.
This paper is a Position Paper. The core of this paper is to indicate a significant yet overlooked aspect of generalization in the NCO community, rather than to propose a novel method.
Our key contributions are as follows:

- First, we identify and explain a significant, yet heretofore overlooked, aspect of generalization: distance metric generalization.

- Second, we critically reassess the commonly adopted training strategy, i.e., Single-Metric Training (SMT).
Our experimental results demonstrate that simply training the model with diverse distance metrics (i.e., Multi-Metric Training (MMT)) can lead to superior generalization compared to SMT.
The proposal of MMT is not intended to emphasize methodological novelty, but rather to illustrate the limitations of SMT as the prevailing training strategy.

We have outlined a direction for future research in the "Limitations and Future Work" section. MMT represents only a preliminary and basic attempt to address the distance metric generalization. Its proposal aims to inspire the development of more sophisticated and efficient approaches, rather than presenting a final solution.

We will revise our manuscript to further clarify our contributions.

**Weakness 2. Insufficient experiments:**

We conduct additional experiments utilizing GOAL [1], a state-of-the-art distance-matrix-based model with publicly available code. Results are reported in the following tables.

Table 1: Performance of the GOAL [1] models on instances with multiple distance metrics.

|                                  | GOAL-SMT | GOAL-MMT |
| -------------------------------- | -------- |--------|
| 2D Euclidean Distance            | 4.33%    | 4.22%  |
| 3D Euclidean Distance            | 4.49%    | 3.20%  |
| 2D Manhattan Distance            | 4.10%    | 4.13%  |
| 3D Manhattan Distance            | 3.95%    | 2.97%  |
| 2D Maximum Distance              | 6.19%    | 4.83%  |
| 3D Maximum Distance              | 5.83%    | 3.79%  |
| Geographical Distance            | 5.07%    | 4.48%  |
| Obstructed 2D Euclidean Distance | 6.15%    | 5.09%  |

Table 2: Performance of the GOAL [1] models on real-world VRP instances.

|                       | GOAL-SMT | GOAL-MDT | GOAL-MDMMT |
| --------------------- | -------- | -------- | ---------- |
| 2D Euclidean Distance (61) | 4.25%    | 3.73%    | 3.70%      |
| 3D Euclidean Distance (2) | 9.10%    | 8.61%    | 8.28%      |
| Geographical Distance (6) | 2.44%    | 2.14%    | 2.00%      |
| EXPLICIT Distance (5)    | 1.82%    | 1.61%    | 0.93%      |
| Google Maps Distance (2)  | 5.51%    | 4.90%    | 4.69%      |
| OSRM Distance (2)        | 6.27%    | 5.27%    | 4.55%      |
| All Instances (78)       | 4.16%    | 3.67%    | 3.56%      |

We train models for 90 epochs. The results reveal that the model trained on instances associated with a single distance metric exhibits a notable performance degradation when evaluated on instances using unseen distance metrics. Simply training model with diverse distance metrics can effective improve the model generalization across different metrics.

---

### Author Response · Authors · 2025-11-25
**General Response [2/2]**

**Question 1. Real-world applications for the Manhattan and Maximum distance metrics:**

We provide the following real-world applications that motivated the inclusion of the Manhattan and Maximum distance:

Manhattan Distance:

- This metric is highly relevant in urban logistics and delivery [2], particularly in cities with grid-based street layouts, such as Manhattan, where vehicles cannot travel directly between points and must traverse horizontal and vertical roads. Furthermore, the Manhattan distance is widely used in chip manufacturing technology. In most cases, chip manufacturing technology only allows horizontal and vertical connections. Thus, the distance between two points in a scan-chain TSP is measured using paths that travel only horizontally or vertically, analogous to walking the streets of Manhattan [3,4].

Maximum Distance:

- A key application of this metric arises in X-ray crystallography experiments [5], where a detector must navigate among diffraction spots in a 3D space parameterized by three angular coordinates $\phi, \chi, \theta$. Since the motors of detectors actuating these angular axes can operate concurrently, the distance between diffraction spots is governed by the slowest motor. Therefore, the maximum component distance metric is appropriate for measuring the distance between diffraction spots.

We will incorporate these specific real-world applications into the revised manuscript.



**References**

[1] Drakulic D, Michel S, Andreoli J M. GOAL: A generalist combinatorial optimization agent learner[J]. arXiv preprint arXiv:2406.15079, 2024.

[2] Kadyrov S, Azamov A, Abdumajitov Y, et al. Deep reinforcement learning for dynamic vehicle routing with demand and traffic uncertainty[J]. Operations Research Perspectives, 2025: 100351.

[3] Cook W J. In pursuit of the traveling salesman: mathematics at the limits of computation[M]. Princeton university press, 2015.

[4] Jünger M, Reinelt G, Rinaldi G. The traveling salesman problem[J]. Handbooks in operations research and management science, 1995, 7: 225-330.

[5] Reinelt G. TSPLIB—A traveling salesman problem library[J]. ORSA journal on computing, 1991, 3(4): 376-384.

---

### Meta-Review · Area_Chair_reeY · 2025-12-30

**Summary:**

This paper presents an empirical study on the generalization of Neural Combinatorial Optimization (NCO) models across different distance metrics for Vehicle Routing Problems. The authors highlighted that the dependency of current NCO methods on a single distance metric restricts their ability to generalize to different distance metrics encountered in real-world scenarios, and introduced a "Multi-Metric Training (MMT)" strategy, which significantly improves generalization across various distance metrics.

All reviewers raised major concerns on the lack of methodological novelty and insufficient experiments. The authors provided detailed responses and added additional experiments. Some concerns can be addressed by additional experiments, but the common concern on the limited novelty of the proposed method remains.

**Reviewer Concerns:**

Reviewer QqGB has major concerns: 1) Profound lack of methodological novelty; 2) underdeveloped motivation for the benchmark; 3) incremental contribution compared to recent SOTA "generalist" NCO solvers; 4) significant confounding variables in the "real-world" experiment; 5) arbitrary design of the MMT training set.

The authors provided detailed responses and added additional experiments. I think most of the concerns of Reviewer QqGB can be addressed by the response, but the concern on the limited methodological novelty remains.

Reviewer BwPX has major concerns: 1) Representativeness of the selected distance metrics; 2) lack of theoretical or interpretive discussion on the performance of different distance metrics; 3) limited novelty of the proposed MMT approach; 4) consider other state-of-the-art distance-matrix-based models as baselines.

The authors provided detailed responses and added additional experiments. I think some of the concerns of Reviewer BwPX can be addressed by the response, but the concern on the limited novelty of the proposed MMT approach remains.

Reviewer RE4j has major concerns on the limited innovation of the proposed method and insufficient experiments.

The authors added additional experiments, which may address the reviewer’s concern on insufficient experiments to some extent. The limited innovation of the proposed method  remains.

**Reviewer Scores:**

I think Reviewer QqGB may not change her/his score, as the concern on the limited methodological novelty remains.

I think Reviewer BwPX may not change her/his score, as the concern on the limited novelty of the proposed MMT approach remains.

I think Reviewer RE4j may not change her/his score, as the limited innovation of the proposed method remains.

---

### Decision · Program_Chairs · 2026-01-26

Reject